# Chronic Effects of a High Sucrose Diet on Murine Gastrointestinal Nutrient Sensor Gene and Protein Expression Levels and Lipid Metabolism

**DOI:** 10.3390/ijms22010137

**Published:** 2020-12-25

**Authors:** Patrick O’Brien, Ge Han, Priya Ganpathy, Shweta Pitre, Yi Zhang, John Ryan, Pei Ying Sim, Scott V. Harding, Robert Gray, Victor R. Preedy, Thomas A. B. Sanders, Christopher P. Corpe

**Affiliations:** 1Nutritional Sciences Division, Faculty of Life Sciences and Medicine, School of Life Courses, King’s College London, Room 3.114, Franklin-Wilkins Building, 150 Stamford Street, London SE1 9NH, UK; p.obrien@gmx.net (P.O.); ge.han@kcl.ac.uk (G.H.); priya.ganpathy@gmail.com (P.G.); shweta.pitre@gmail.com (S.P.); zoeyzhang9496@outlook.com (Y.Z.); j_h_ryan@hotmail.com (J.R.); spying22@gmail.com (P.Y.S.); robert.gray@kcl.ac.uk (R.G.); victor.preedy@kcl.ac.uk (V.R.P.); tom.sanders@kcl.ac.uk (T.A.B.S.); 2Department of Biochemistry, Memorial University, Elizabeth Avenue, St. John’s, NL A1C5S7, Canada; h32svh@mun.ca

**Keywords:** gastrointestinal, sucrose, nutrient sensors and transporters, gut peptides, nitric oxide

## Abstract

The gastrointestinal tract (GIT) plays a key role in regulating nutrient metabolism and appetite responses. This study aimed to identify changes in the GIT that are important in the development of diet related obesity and diabetes. GIT samples were obtained from C57BL/6J male mice chronically fed a control diet or a high sucrose diet (HSD) and analysed for changes in gene, protein and metabolite levels. In HSD mice, GIT expression levels of fat oxidation genes were reduced, and increased *de novo* lipogenesis was evident in ileum. Gene expression levels of the putative sugar sensor, *slc5a4a* and *slc5a4b*, and fat sensor, *cd36*, were downregulated in the small intestines of HSD mice. In HSD mice, there was also evidence of bacterial overgrowth and a lipopolysaccharide activated inflammatory pathway involving inducible nitric oxide synthase (iNOS). In Caco-2 cells, sucrose significantly increased the expression levels of the *nos2*, iNOS and nitric oxide (NO) gas levels. In conclusion, sucrose fed induced obesity/diabetes is associated with changes in GI macronutrient sensing, appetite regulation and nutrient metabolism and intestinal microflora. These may be important drivers, and thus therapeutic targets, of diet-related metabolic disease.

## 1. Introduction

The prevalence of obesity and associated co-morbidities, such as type 2 diabetes (T2DM) has substantially increased over the last 4–5 decades [1], and it has been estimated that between 6% (Japan) and >30% (USA) of the world’s population are now obese [2]. As per the International Diabetes Federation, 463 million adults (20–79 years) were living with diabetes in 2019 and the number is expected to rise to 700 million by 2045. Associated with these metabolic derangements are an increased risk of non-alcoholic fatty liver disease (NAFLD) hypertension and cardiovascular disease (CVD) leading to a reduction in quality of life and premature death. Management of these diseases remains a challenge because of the prevailing obesogenic environment in both Western and developing countries. Increased availability of palatable, energy-dense foods, genetic predisposition and a sedentary lifestyle are all critical factors; yet the specific mechanisms of diet-related metabolic disease are not fully understood.

Sucrose is abundant in many processed foods and drinks, including sugar-sweetened beverages (SSBs). In more recent years, consumption of SSB has dramatically increased in children and young adults [3,4] contributing to excess calorie intake and obesity while providing minimal nutritional value [5]. Systematic reviews and meta-analyses have identified an association between SSBs intake and the development of obesity and diabetes [6,7], although there are conflicting findings in the literature [8,9]. Despite these uncertainties, in 2015, the World Health Organization (WHO) and Scientific Advisory Committee for Nutrition (SACN) recommended restricting added sugar intake to between 5–10% of daily calories [10,11]. If added sugars, principally in the form of sucrose, are causal in the development of metabolic disease, understanding the mechanisms of action could further support current recommendations and potentially lead to the identification of new therapeutic targets and treatments.

The gastrointestinal tract (GIT) plays an important role in regulating nutrient metabolism and energy intake [12]. When food is ingested and digestion begins, taste cells on the tongue and entero-endocrine cells distributed along the length of the GIT sense the ingested sugars, fats and proteins and release a number of signalling peptides (e.g., glucose-dependent insulinotropic polypeptide (GIP) glucagon-like peptide-1/(GLP-1/2) cholecystokinin (CCK) and peptide YY (PPY)) that have both endocrine and paracrine effects [13]. These peptides influence the function of the GIT (e.g., intestinal secretions, gastric emptying and nutrient absorption) nutrient metabolism (e.g., insulin secretion) and food/energy intake. Meta-analysis of human T2DM data suggests minimal differences in circulating incretins compared to healthy people [14,15]. However, macronutrient sensing has been shown to be dysfunctional in animal models of obesity and diabetes [16,17]. For example, in rodents fed a high-fat diet, duodenal nutrient-sensing mechanisms are disrupted, affecting glucose homeostasis [18]. Small intestinal bacterial overgrowth (SIBO) is also a potential causal factor for metabolic disorders including obesity, diabetes and NAFLD [19]. In obese patients, the prevalence of SIBO is as high as 41% [20]. The underlying mechanism remains ill-defined, but it has been hypothesized that an imbalance in available and unavailable carbohydrate levels may promote gut microbiota dysbiosis and overgrowth [21]. Lipopolysaccharide (LPS) found on the outer membrane of Gram-negative bacteria also increases nitric oxide (NO) synthesis by pro-inflammatory cytokines in obesity-induced chronic inflammation [22,23]. NO is a labile lipid soluble gas and an important signalling molecule, participating in many physiological processes such as gastrointestinal homeostasis, inflammation and the maintenance of vascular tone [24,25,26]. Several studies have reported that NO is also a sweet taste-signalling molecule [27] that can regulate food intake [28]. Understanding the pathways by which dietary changes result in metabolic disease via alteration in gut microbiota, inflammation and nutrient sensing are therefore of interest [29].

To identify novel pathways in the development of metabolic disease, we undertook transcriptomic, proteomic and metabolomic analysis of GIT samples obtained from mice chronically fed a high sucrose diet for 6 weeks. We also used Caco-2 cells as an in vitro model of the enterocyte to support our in vivo findings. 

## 2. Results

### 2.1. Dietary Intake, Anthropometry and Blood Biochemistry Analysis of Control and Sucrose Fed Mice 

Control fed mice had free access to water and the control pellet chow, and high sucrose diet (HSD) fed mice had free access to water, the control pellet chow and light condensed milk (LCM). As shown in Figure 1a, HSD fed mice almost completely neglected the control pellet chow. Adult mice normally consume 3–5 g of chow/day (12 g chow/100 g body weight). Daily average intake of the control fed group was 3.15 ± 0.07 g control diet/day compared with 4.88 ± 0.06 g of LCM/day for the HSD group (Figure 1b).

Consumption of LCM altered the macronutrient intake of HSD fed mice when compared to controls (Figure 1c). Mice in the HSD group had a significantly higher intake of total carbohydrates, mainly sucrose and lactose when compared to the controls. HSD mice derived 62% of their total caloric intake from sucrose, compared to <11% in the controls (*p* < 0.001). Fat consumption at 5% of total daily calories was comparable between control and HSD mice, but protein intake was significantly lower in HSD mice when compared to controls (24%, control vs. 14%, HSD). The control diet (AIN93G) contained approximately 4.7% of dietary fibre, and based on an average intake of 3.15 g control pellet/day for controls and 0.2 g control pellet/day for HSD mice, we estimated that fibre intake for controls was 0.16 g/day and for HSD mice it was 0.01 g/day.

The increased food intake in HSD fed mice resulted in a higher energy intake over the course of the study (Figure 1d). Mean daily calorie intake of the control group was 11.56 ± 0.15 compared with 13.42 ± 0.16 kcal/day (*p* ≤ 0.001) for the HSD group (Figure 1e). Body weights were also 22% higher in the HSD fed mice compared to controls (Figure 1f). At 6 weeks, the average body weights of the control group were 20.6 ± 0.66 g compared with 25.2 ± 0.42 g for the HSD group. 

Biochemical analysis of serum showed that mean fasting glucose levels were significantly higher in HSD mice when compared to controls: 5.23 mmol/L for the control group vs. 11.89 mmol/L for the HSD group (Figure 1g). HSD fed mice were not hyperinsulinemic compared to controls and blood lipids were not different. Serum cholesterol approached significance (1.82 vs. 2.08 mmol/L, *p* = 0.06, HSD vs. control) while triglycerides (0.71 vs. 0.48 mmol/L; *p* = 0.7) and NEFA (1.50 vs. 0.85 mmol/L; *p* = 0.31) were clearly comparable between treatments (HSD vs. control, respectively). 

### 2.2. qPCR Analysis of Small Intestinal Fat Oxidation Genes 

Expression levels of fat oxidation genes were also studied in the GIT of control fed and HSD mice (Figure 2). In control fed mice, *Pdk4* gene expression levels were highest in tongue and jejunum (Figure 2a). *Pdk4* gene expression levels were significantly downregulated in all intestinal segments obtained from HSD mice when compared to control fed mice (Figure 2b). Similarly, jejunal *hmgcs2, acot1, acot2, me1* and *cyp4a10* gene expression levels were all significantly downregulated in HSD mice when compared to control fed mice (Figure 2c–g). Fat oxidation genes were also significantly downregulated in the duodenum and ileum of HSD mice (data not shown). Array data indicated fat synthesis gene expression levels were unchanged in small intestinal segments (data not shown).

### 2.3. Analysis of Intestinal Lipid Metabolites 

Analysis of the fatty acid composition of all intestinal tissues was undertaken; however, we only found significant differences in the ileum (Figure 3a). The HSD treatment resulted in higher levels of 16:0, 16:1 and 18:1n-9 and lower levels of 14:0, 18:0, 18:2n-6, 20:3n-6 and 20:4n-3 in ileum compared to controls. The 16:0/18:2n-6 ratio, serving as an indicator of *de novo* lipogenesis (DNL) was also increased in ileal samples obtained from rodents fed the HSD diet (Figure 3b). The 18:1n-9/18:0 ratio, serving as an indicator for stearoyl-CoA desaturase (SCD) activity, was also studied using the data from lipid analysis (Figure 3c). The 18:1n-9/18:0 ratio in ileum was found to be significantly higher in HSD fed mice compared to controls. Liquid chromatography–mass spectrometry was performed to quantify oleoylethanolamide (OEA) and palmitoylethanolamide (PEA), each derived from 18:1n-9 and 16:0, respectively. Similar to the SCD desaturase activity, the OEA:PEA ratio was significantly higher in the HSD group, but only in ileum (Figure 3d).

### 2.4. Validation of Microarray Data Obtained from the Small Intestines of Control and Sucrose Fed Mice

From the gene array data, taste- and nutrient-sensing genes including fat-, umami-, sour- and bitter-taste genes, as well as downstream signalling molecules, such as Gustducin, were manually extracted and fold changes reported (Table 1). Only minor changes in intestinal nutrient sensor gene expression levels were detected; however, the gene expression levels for the Na^+^ dependent sugar sensor SGLT3 (*slc5a4a* and *slc5a4b*) and the fat sensor CD36 (*cd36*) were downregulated by as much as 4-fold in the duodenum and jejunum of HSD fed mice.

To validate the array data, qPCR analysis of sugar sensor genes expression levels was undertaken in tongue, stomach, duodenum, jejunum and ileum tissues obtained from control and HSD fed mice (Figure 4). Consistent with array data, *Tas1r2* was unchanged in control and HSD fed mice in all tissues (Figure 4a). *Tas1r3* was also unchanged in control and HSD fed mice along the length of the GIT, although a modest (28%) downregulation in jejunum (*p* < 0.05) and ileum (*p* = 0.08) was noted in HSD fed mice (Figure 4b). SGLT3a (*slc5a4a*) and SGLT3b (*slc5a4b*) were both downregulated in the duodenum and jejunum (Figure 4c,d) as was *cd36* (Figure 4e), although significance was only reached in the jejunal samples.

### 2.5. Western Blot Analysis of Small Intestinal Sugar Sensors and Transporters

To assess changes in GLUT2 and SGLT3 protein expression levels, we undertook Western blot analysis of duodenal, jejunal and ileal proteins harvested from control and HSD fed mice. Western blot analysis showed no change in SGLT3 (Figure 5a–c) in duodenum, jejunum and ileum obtained from control and HSD mice.

GLUT2 protein expression levels were also unchanged in duodenum and jejunum (Figure 5a,b), but significantly increased 1.5-fold in ileum (*p* < 0.01) (Figure 5c), consistent with the pattern of GLUT2 gene expression changes detected in array data analysis of pooled RNA samples. In arrays, sucrase- Isomaltose and SGLT1 gene expression levels were also unchanged in all small intestinal regions, whereas GLUT5 was downregulated almost 2-fold but only in the ileum (data not shown).

### 2.6. qPCR Analysis of Gut Peptide Gene Expression Levels

Gene expression levels of ghrelin (*ghrl*), CCK (*cck*) and preproglucagon (*gcg*) in intestinal segments of control and HSD fed mice were also analysed by qPCR (Figure 6). *Ghrl* showed a 4-fold induction in the stomach (Figure 6a) but did not reach statistical significance (*p* = 0.13). *Cck* expression was significantly upregulated by 65% in the jejunum of HSD mice when compared to controls (Figure 6b) but not in duodenum and ileum. Expression of *gcg*, the gene responsible for the synthesis of GLP-1 and -2, was increased in the ileum by 65% (*p* = 0.055) (Figure 6c) and decreased in the duodenum by 50% (*p* = 0.11) and was unchanged in jejunum. Array data indicated that gene expression levels for ghrelin (duo: −1.52, jej: 1.03, ile: 1.21) and gip (duo: −1.20, jej: −1.16, ile: 1.21) were minimally changed in all three small intestinal segments obtained from control and HSD fed mice (data not shown).

### 2.7. Identification of an LPS Mediated Inflammatory Pathway by GeneGo Metacore Pathway Analysis of Microarray Data and Validation by qPCR

Pathway analysis of our array data using GeneGo MetaCore software identified lipopolysaccharide (LPS) induced inflammation in the small intestines of HSD fed mice (Figure 7). LPS triggers an inflammatory pathway modulated by macrophage inhibitory factor (MIF), which promotes increased gene expression of myeloid differentiation protein-2 (MD-2), inhibitory factor KB (I-kb), inducible NOS and interleukin-1, summarized in Table 2.

To validate the array data, qPCR analysis of inflammatory pathway genes was undertaken. *LY96* was upregulated 2–2.5-fold in jejunum and ileum (Figure 8a), and NFKb was upregulated 2-fold in duodenum (Figure 8b) but did not reach significance. *IL-1* was upregulated 3-fold in jejunum (*p* < 0.05) (Figure 8c). qPCR analysis of *NOS2* along the length of the GIT of control fed mice showed the highest expression levels in the distal regions of the small intestine (Figure 8d). Consistent with the array data and pathway analysis, qPCR studies showed that *NOS2* gene expression levels in HSD mice were upregulated in the duodenum and jejunum (Figure 8e) as well as all other intestinal tissues when compared to control fed mice. *NOS2* was significantly upregulated by 7-fold in the tongue (*p* < 0.001), 34-fold in the duodenum (*p* < 0.05), 6-fold in the jejunum (*p* < 0.05) and 4-fold in the ileum (*p* = 0.055). In the stomach, *NOS2* was upregulated 2-fold in HSD mice but did not reach statistical significance.

### 2.8. qPCR of the Small Intestinal Microflora

To assess changes in the microbial population along the length of the GIT in control and HSD mice, qPCR analysis of *16S* rRNA (a bacterial marker gene) was undertaken (Figure 9). In control fed mice, tongue, stomach and ileum had the highest *16S* rRNA expression levels and duodenum and jejunum the lowest (Figure 9a). Following exposure to HSD, there was a 4–12-fold upregulation in 16s rRNA expression levels in the stomach (*p* < 0.05), duodenum and ileum (Figure 9b). No significant fold changes in *16S* rRNA gene expression were observed in the tongue and jejunum (*p* > 0.05).

### 2.9. The Effects of Sugars and LPS on Nitric Oxide Production In Vitro

To study the regulation of gastrointestinal NO by sugars and microbiota, we exposed Caco-2 cells (an in vitro model of the human enterocyte) to glucose, sucrose and LPS for 48 h and measured NOS2 gene and iNOS protein expression levels and NO gas secretion (Figure 10). When compared to 5 mM glucose, NOS2 mRNA levels were upregulated by 25 mM sucrose and 100 µg/mL LPS with 25 mM sucrose (*p* < 0.05) (Figure 10a) but not by 25 mM glucose. iNOS protein is a 1135 amino acid protein with a predicted Mw of 130 kD [30]. However, a single 50 kD iNOS protein was detected when Caco-2 cell samples were Western blotted (Figure 10b). Furthermore, when compared to the glucose control, a 48 h exposure to 25 mM sucrose resulted in a significant increase in iNOS protein expression levels, whereas protein expression levels were unresponsive to 25 mM glucose or 1 µg/mL LPS with 25 mM sucrose. When compared to 5 mM glucose control, NO gas production was significantly increased after Caco-2 cells were exposed to 25 mM sucrose (*p* = 0.004), and 100 µg/mL LPS with 25 mM sucrose (*p* = 0.001) for 48 h, but not following exposure to 25mM glucose (Figure 10c).

## 3. Discussion

The aim of this study was to identify changes in GIT function in mice chronically fed a high sucrose diet to induce obesity and diabetes. In the present study, C57BL/6 mice were given free access to a control pellet diet containing 2% corn oil and sweetened light condensed milk (LCM) supplemented with 1% corn oil. Animals consumed significantly higher amounts (*p* < 0.001) of LCM when compared to the control pellet diet, resulting in a switch from dietary starch to added sugars (LCM, 45% sucrose/ 10% lactose). Associated with this dietary switch was an increase in energy intake (hyperphagia) (*p* < 0.001) most likely driven by increased palatability of the LCM diet via stimulation of dopamine in the mesolimbic system and the hypothalamic reward centres [31,32]. The observed hyperphagia is unlikely to be responsible for the observed weight gain and metabolic disturbances, since pair fed studies using HSD and control fed rodents also reported weight gain, diabetes and hepatic steatosis [33]. Consistent with other reports, the HSD fed mice were also hyperglycaemic, but not hyperinsulinemic, suggesting the animal was diabetic due to insulin resistance and pancreatic beta cell exhaustion [34,35,36].

When dietary sucrose is ingested, it is split into free glucose and fructose in the small intestine by sucrase isomaltose (SI) expressed on the apical membrane of enterocytes. The free glucose and fructose is then transported across the gut via SGLT1 and GLUT5 expressed on the apical membrane and GLUT2 expressed on the basolateral membrane [37]. Approximately 25% of the glucose load that enters the portal vein is extracted by the liver, whereas almost all of the fructose is extracted by the liver during the first pass [38]. In addition, because fructose metabolism in the liver bypasses key rate limiting glycolytic enzymes (e.g., phosphofructokinase (PFK)), carbons derived from fructose that are in excess of energy needs of the liver are converted into glucose and lactate via gluconeogenesis, and fatty acids via DNL. Increased DNL involves not only enhanced fat synthesis but also suppression of fat oxidation [39]. Triglycerides produced via DNL are either stored in the liver or exported into blood raising circulating triglyceride levels [40]. The idea that the liver is the principal site of fructose metabolism has, however, recently been questioned [41]. In mice studies, when fructose loads are low (0.25–0.5 g/kg body weight), the gut converts the sugar into glucose and organic acids, such as fatty acids, uric acid, glycerate and glutamate. When loads of fructose are high (>1 g/kg body weight), the capacity of the gut to metabolise fructose is exceeded and the free fructose passes to the liver where it is then metabolised as described above. In addition, in Chinese hamsters fed a high fructose diet (60% *w*/*w*) for three weeks, lipid synthesis was enhanced in enterocytes, resulting in increased synthesis of apoB48 and release of intestinal-derived chylomicrons, likely via activation of the carbohydrate response element binding protein (ChREBP) and the facilitative fructose transporter, GLUT5 [42]. In small intestinal segments of HSD mice, fatty acid oxidation (FAO) genes were significantly decreased, whereas fatty acid synthesis genes were unchanged (array data not shown) when compared to controls. Although the decrease in FAO gene expression suggests the balance in lipid metabolism in the gut has shifted toward lipid synthesis, evidence of increased DNL was detected only in ileal samples of HSD mice. In addition, although rodent studies have shown that the intestinal contribution to circulating triglycerides ranges between 10 to 40% of total plasma triglyceride [43], the increased ileal DNL in HSD mice was not sufficient to raise blood triglycerides. The liver is the principle tissue responsible for fructose induced hypertriglyceridemia, and long-term exposure to dietary sucrose (and fructose) results in the development of fatty livers (NAFLD) which can progress to non-alcoholic steatohepatitis (NASH) [44]. NAFLD/NASH is associated with a decrease in the secretion of TGs, synthesis of apolipoprotein B-100 and import and export of VLDLs [45], which may also explain why circulating triglycerides in HSD mice were no different to controls. 

The OEA:PEA ratio was increased in HSD mice ileum. OEA is a fatty acid ethanolamine of oleic acid that is synthesized by proximal enterocytes upon feeding [46] and induces satiety via activation of the transcriptional factor, PPAR-alpha, which suppresses *iNOS* gene expression [47,48]. *iNOS* gene expression levels were, however, increased in all regions of the GIT, including the ileum, suggesting the satiating effect of ileal OEA was overwhelmed by other metabolic events discussed below.

Distributed along the length of the GIT are numerous nutrient (energy) sensing cells (e.g., taste cells, X/A cells, K, I and L cells) that release a number of signalling peptides (e.g., ghrelin, GIP, CCK and GLP1) in response to the ingestion of fats, carbohydrates and proteins [49,50]. The signalling peptides have both paracrine and endocrine effects, influencing taste preference, gastric emptying, nutrient absorption, insulin secretion and food intake. The molecular mechanisms by which macronutrients are sensed by gut have largely been established (e.g., the sweet taste receptors: T1r2/3, SGLT1 and SGLT3; fat sensors: CD36 and GPR120; and amino acid sensors, PEPT1 and LAT1) [51,52,53]. Studies have also shown that when the luminal macronutrient load changes, there are alterations in nutrient sensor gene expression levels and nutrient perception [54,55]. The HSD mice showed a clear preference for the sucrose sweetened milk in our study; however, *tas1r2/3* gene expression levels along the length of the GIT tract were unchanged compared to controls, suggesting that changes in sweet taste preference are not mediated via alterations in *tas1r2/3* receptor gene expression levels. Alternative sugar sensor mechanisms exist on the tongue, such as amylase, SI and SGLT1 [56], and we cannot rule out changes in their expression levels altering taste sensitivity. In the duodenum and jejunum of HSD fed mice, we did detect a downregulation in the gene expression levels of the putative sugar sensor SGLT3, namely, *slc5a4a* and *slc5a4b*. SGLT3 is expressed in cholinergic neurones in the submucosal and myenteric plexus and in portal vein, and is thought to be a sugar sensor that activates vagal afferents [52]. Because SGLT3 activity is sensitive to pH, SGLT3 may also play a role in regulating gastric emptying. Because SGLT3 protein expression was unchanged in HSD mice compared to controls, it is difficult to conclude what the physiologic importance of altered SGLT3 gene expression levels are. Discordance in gastro-intestinal SGLT3 gene and protein expression levels has previously been detected [57] suggesting post-translational modification. We speculate that increased levels of dietary simple sugars may stabilize the protein product, and thus gene transcriptional activity required to maintain protein expression levels are reduced. Taste sensitivity to fat is also influenced by the levels of dietary fat, that is, low fat diets result in increased sensitivity to fat and vice versa [58]. Furthermore, sensitivity appears to be mediated by alterations in the expression levels of the fat sensor, CD36 [59,60]. In the HSD mouse, *cd36* gene expression levels were also significantly downregulated in duodenal and jejunal samples, suggesting that sensitivity to dietary fat could be lower in the HSD mice. CD36 plays an important role in upregulating fat oxidation, and blocking CD36 inhibits fatty acid oxidation and promotes fat accumulation [61]. Although fat intake levels in the control and HSD diet were unchanged, these data suggest a novel pathway by which added sugars may result in overconsumption, that is, via a reduced ability to sense and respond to dietary fat. Future studies in rodents and people could test this hypothesis.

Because HSD mice were hyperphagic we also assessed the gene expression levels of the anorexigenic/ orexigenic and incretin hormones in our intestinal samples. In HSD mice, the gene expression levels of the hunger hormone, *ghrl* (Ghrelin), were upregulated four-fold in stomach, although it did not reach statistical significance. The detected increase in ghrelin gene expression levels might be due to an increase in the number of ghrelin positive X/A cells, as this has been observed in the stomachs of obese mice fed a high fat diet [62]. Together, these data suggest that the increase in the number of ghrelin producing cells in obesity might be due to a systemic signal altering the cellular repertoire of the stomach, rather than alterations in luminal macronutrient levels. In humans who are obese and have T2DM, there is evidence of ghrelin resistance, and ghrelin secretion and levels in blood are in fact decreased [63,64]. The increase in ghrelin gene expression (via increased ghrelin positive cell number) may, therefore, be an attempt by the stomach to compensate for a reduction in ghrelin secretory capacity [62]. 

In HSD mice, *cck* gene expression levels were upregulated only in the jejunum. CCK is a satiety hormone synthesised in I cells that is released in response to dietary fatty acid and amino acid intake inhibiting gastric emptying via activation of vagal afferents [65]. Elevations in CCK levels in the small intestines of obese humans have been detected, and it has been suggested that this may be an adaptative response to CCK-receptor resistance [66,67]. CCK is known to suppress carbohydrate intake and reduce food intake via the CCK-A receptor [68]. Our finding suggests that the gut compensates for reduced CCK signalling by increasing *cck* gene expression, either via changes in transcription or I cell number.

In HSD mice, gene expression levels of *gcg*, the precursor of GLP1/2, were downregulated in duodenum and upregulated in ileum, although the fold changes did not quite reach statistical significance. GCG is expressed in L cells, and these data may reflect an alteration in either gene transcription or L cell density along the length of the GIT in HSD mice, which has previously been observed in the large intestines of high fat fed models of obesity and in diabetes, although not in the small intestine [69]. In diabetes, GLP1 secretion is only mildly attenuated [70]. The impairment in GLP1 secretion may occur primarily in the duodenum and be responsible for the proposed anti-incretin effect detected in morbidly obese individuals with T2DM [71]. Our data suggest that this could be partially compensated for by increased *gcg* expression and GLP1 secretory capacity in the distal gut. 

To further investigate changes in GIT gene expression levels, GeneGo MetaCore pathway analysis of small intestinal segments obtained from control and HSD mice was undertaken which identified a LPS activated inflammatory pathway involving iNOS, which we validated by qPCR. LPS is found in the outer membrane of Gram-negative bacteria, and previous studies have shown the Gram-negative microbiota is elevated in obese patients when compared to healthy people [72]. Small intestinal bacterial overgrowth (SIBO) is characterized by excessive growth of bacteria in the small intestine, and is cited as a potential causal factor for NAFLD diseases and metabolic disorders including obesity and diabetes [19]. SIBO was observed in 41% of obese patients [73]. Moreover, it was found that 75% of diabetic patients with SIBO showed improved GIT symptoms after treatment with antibiotics [74]. The underlying mechanisms linking SIBO and metabolic disorders remain ill-defined, but it has been hypothesized that alterations in available and unavailable dietary carbohydrate levels may promote gut microbiota dysbiosis and overgrowth [21]. Indeed, in our HSD mouse, sucrose replaced starch as the principle ingested available carbohydrate, and fibre intakes were dramatically reduced in HSD mice because of the switch from the control pellet to LCM. This raises the possibility that in a low fibre environment, the intestinal microbiota migrate out of the large intestine in search of energy providing substrates, and in the proximal regions of the gut, they may compete with the energy sensor cells of the gut for substrates, potentially leading to disturbances in nutrient sensing, peptide secretion, metabolism and even behaviour [75]. In addition, when LPS is synthesized by intestinal bacteria, it enters the host bloodstream and promotes the upregulation of pro-inflammatory cytokine expression [76,77]. In HSD mice, there is also evidence that LPS upregulates several genes involved in intestinal inflammation. LPS and inflammatory molecules such as Il-1β downregulate GLUT5 [78,79], which might explain the reduction in GLUT5 gene expression in HSD ileum (array data not shown). The detected change in iNOS is also associated with the pathogenesis of metabolic diseases. Several studies have shown NO to be an important GIT signalling molecule and that inhibition of NOS by N omega-Nitro-L-arginine methyl ester hydrochloride (L-NAME) suppresses appetite and decreases intake of a palatable milk diet [80,81].

In HSD mice, *16sRNA* gene levels were increased in all regions of the GIT, except for the jejunum. Therefore, the detected changes in jejunal gene expression levels, including the upregulation in *nos2,* could not have been driven by changes in bacterial load/LPS. Indeed, in our Caco-2 cell experiments, we showed that sucrose can also cause an increase in *nos2* gene, iNOS protein and NO gas production. The genes responsible for the sweet taste receptor homodimer T1R3 and the putative sugar sensor SGLT3 are expressed in Caco-2 cells [82]; however, it is unlikely that sucrose’s effect on iNOS is mediated by the sugar sensors, simply because T1R3 responds only to very high levels of sucrose (>300–500 mM) [83] and depolarisation of the cell membranes occurs only when glucose and its analogs bind to SGLT3 [52]. When dietary sucrose is ingested, it can be partially broken down into free glucose and fructose by SI expressed on the taste cells of tongue and in small intestine and by the acid present in the stomach. SI is also expressed in Caco-2 cells [84]. Glucose had no effect on NO in Caco-2 cells, suggesting fructose, along with LPS mediated inflammation, is most likely responsible for the changes in NO. In support, in fructose fed rodents, the levels of blood NO are increased [85] and iNOS gene, and protein expression levels are increased in the aortas of high fructose corn syrup (HFCS) fed rodents [86], although endothelial eNOS is decreased resulting in dysregulation of vascular tone and hypertension [87].

The molecular mechanisms by which dietary sucrose ( and fructose) results in the detected change in gut gene expression levels remain unclear. From our array data, we did not find any changes in the expression levels of transcriptional factors involved in regulating sugar transport and metabolism, namely, ChREBP [88] thioredoxin interacting protein (TxNip) [89] or Liver X receptor-α (LXR-α) [90]. Increased intracellular fructose leads to elevations in uric acid levels which might be important [39]. Uric acid can activate NADPH oxidase, leading to oxidative stress and the activation of the transcription factor, NFkB, which in turn can lead to increased *nos2* gene transcription. Mitochondrial oxidative stress can also cause a reduction in aconitase-2 activity [91] and enoyl CoA hydratase-1, a rate-limiting enzyme in β-fatty acid oxidation [92]. Uric acid is also known to increase AMP Deaminase activity which blocks fat oxidation by inhibiting the energy sensor AMP-activated protein kinase [93]. Future studies will, therefore, involve studying the regulation of intestinal nutrient sensors, SGLT3 and CD36, by uric acid and its downstream signalling molecules.

In summary, dietary sucrose induced obesity/diabetes in mice is associated with alterations in numerous gastrointestinal genes involved in macronutrient sensing, appetite regulation, nutrient metabolism and intestinal microflora. Causality, however, remains an unanswered question. To better understand the importance of the changes in microflora and iNOS in intestinal nutrient sensing and the HSD phenotype, in the future, a more detailed characterization of the microflora, probiotic and prebiotic interventions and treatment with the nitric oxide synthase inhibitor (L-NG-nitro-arginine methyl ester) will be undertaken. These studies will identify viable therapeutic targets and strategies for the management of diet related metabolic disease.

## 4. Materials and Methods

### 4.1. Laboratory Animals and Study Protocol

Sixteen C57BL/6J juvenile male mice (28 days of age) were obtained from Charles River, UK, and were randomly assigned to two groups (*n* = 8 each): a control group or a high-sucrose diet (HSD) group. Mice weighing 16.5 ± 0.59 g and 16.5 ± 0.39 g were placed in the control group and the HSD group, respectively. Mice were individually housed in cages with wood-chip bedding (Utemp1284 cages, Techniplast, UK; Aspen-wood chips, B and K Ltd.) under controlled temperature (20–22 °C) and humidity conditions (45–65%) with a 12 h light/dark cycle. 

In pilot feeding studies, HSD fed mice consumed large amounts of LCM and very little of the control chow diet resulting in changes in sugar as well as fat intakes. In order to ensure fat consumption (specifically linoleic acid) was similar between control and HSD mice, 1% corn oil was added to LCM and 2% corn oil to the control chow diet. Mice were acclimatised for one week with free access to the control diet (2% corn oil pellets) and water. After the acclimatisation period, the control mice were continued on the control diet, and the HSD mice were given free access to light condensed milk (LCM) (Carnation milk, Nestlé, UK) containing 1% corn oil in addition to the control diet (AIN93G) for a period of six weeks (Table 3 for diet composition).

Throughout the intervention, body weights, chow intake and LCM consumption were recorded daily. These data were used to calculate macronutrient and energy intake. After the six-week dietary intervention, all animals were fasted overnight to minimise the variation in the time between the last food intake and blood and tissue collection. All animals were euthanised by cervical dislocation whilst under halothane anaesthesia. Tissues were dissected into tubes and immediately snap-frozen in liquid nitrogen for further RNA and protein extraction. All procedures were carried out according to the project license guidelines under the UK Home Office Animals Scientific Procedures Act, 1986.

### 4.2. Serum Analysis

Blood samples were collected via decapitation and left to drain into a 2 mL sterile tube and whole blood was processed to obtain serum. Serum glucose, insulin, cholesterol and triglycerides were analysed using Siemens Healthcare Diagnostics Ltd. kits, following the manufacturer’s instructions at KingsPath King’s College London Hospital. Non-esterified fatty acids (NEFA) were analysed using a using a calorimetric assay (Wako Diagnostics, Richmond, VA, USA).

### 4.3. Cell Culture

Caco-2 cells were grown in 25 mM D-Glucose Dulbecco’s modified Eagle’s medium (DMEM) containing 10% fetal bovine serum, 50 units penicillin and 50 µg streptomycin, 0.1 mM MEM non-essential amino acids and additional 2 mM L-glutamine (Gibco). Cells were maintained in T25 flasks, split 1:20 once a week at ~80% confluence and seeded for experimentation at a density of 10,000 cells/cm^2^ in 24 multi-well dishes. Medium was changed every other day until confluent, and daily after reaching confluence. For experiments, cell passages between 38 and 50 were used.

Confluent cell monolayers were washed twice with 1X phosphate buffered saline (PBS, G-Biosciences, St. Louis, MO, USA). Cells were then exposed to different concentrations of sugars and LPS including 5 mM glucose, 25 mM glucose, 25 mM sucrose (Sigma-Aldrich, Gillingham, UK), and 100 µg/mL LPS with 25 mM sucrose, separately. LPS was isolated from Escherichia coli (EMD Millipore Corporation, Temecula, CA, USA). After sugar and LPS exposure for 48 h, cells were harvested for RNA and protein extraction, and supernatants collected.

### 4.4. RNA Extraction

Intestinal tissues were incubated overnight in RNAlater^®^-ICE solution (Life Technologies, Carlsbad, CA, USA) at −80 °C, and 15 mg of the tissue was homogenized in Qiazol lysis reagent (Qiagen) in the Qiagen TissueLyser for 3 × 30 s at 30 Hz, and RNA extracted using a RNeasy Lipid Tissue Mini Procedure (Qiagen, The Netherlands), following the manufacturer’s protocol. 

Confluent Caco-2 cells were washed with ice-cold PBS and lysed with cold TRIzol^®^ reagent. Chloroform and RNase-free water were added to the cell lysate, rigorously vortexed for 15 s and incubated for 10 mins at room temperature. Samples were then centrifuged at 13,000 RPM at 4 °C for 10 mins, and the aqueous phase was removed and mixed with ice-cold isopropanol and incubated at −20 °C for 30 min. RNA was precipitated by centrifugation at 13,000 RPM at 4°C for 30 min. The resulting pellet was washed twice in 75% *v*/*v* ethanol and centrifuged for 5 min at 13,000 RPM at 4 °C. The RNA pellet was then re-suspended in RNAse-free water, and RNA concentration (ng/µL) and purity (260/280) were measured (Nanodrop 2000).

### 4.5. Microarray Analysis

An amount of 400 ng of RNA was pooled together from duodenal, jejunal and ileal samples from control and HSD fed mice and RNA integrity confirmed on an Agilent Bioanalyzer chip. RNA was then labelled with a GeneChip^®^ 3′ IVT Express Kit (Affymetrix, Santa Clara, CA, USA) and analysed using the GeneChip MouseGenome 430A 2.0 Array following the manufacturer’s instructions. Data were analysed using the GeneGo software (MetaCore™, version 6.19 build 65960—Thomson Reuters).

### 4.6. Quantitative Polymerase Chain Reaction (qPCR)

An amount of 1 µg of RNA obtained for intestinal tissue segments or Caco-2 cells was reverse transcribed with a high capacity RNA-to-cDNA kit (Applied Biosystems, Waltham, MA, USA) according to the manufacturer’s protocol in a PTC-200 thermal cycler. The resulting cDNA was diluted 1:10 in water, and PCR performed using 10 ng cDNA and a primer concentration of 900 nM with FastStart Universal Probe Master (Rox) (Roche, Switzerland) mastermix. PCR conditions: using AB 7000 qPCR cycler (Applied Biosystems, USA), 10 min at 95 °C for polymerase activation, 40 cycles of 95 °C annealing for 15 s and 1 min at 60 °C for amplification. Primers (Table 4) were designed using Roche’s online ProbeFinder Software, v. 2.50. The mouse genome was selected to design PCR assays over exon–exon borders for transcriptome specificity. All primers were obtained from Integrated DNA Technologies (Leuven, Belgium) and reconstituted with ultrapure water to a stock concentration of 10 μM.

### 4.7. Protein Extractions

An amount of 15 mg of intestinal tissue was dissected on ice and homogenised in ice cold protein extraction buffer (1 mM fresh DTT, 50 mM Tris-HCl pH 7.4, 250 mM mannitol, 100 mM NaCl, 1mM EDTA, 1 mM EGTA, 10% glycerol) containing 1x protease-inhibitor cocktail (Sigma-Aldrich P8340, Dorset, UK) using a Qiagen TissueLyser for 2 × 60 s at 30 Hz. Samples were then incubated on a rotating spinner for 30 min at 4 °C and then centrifuged at 13,000 RPM for 10 min at 4 °C. The protein-containing supernatant was transferred into a new tube. Protein content was quantified using a Pierce™ BCA Protein assay kit.

Caco-2 cell proteins were collected after being washed with 1X cold PBS twice and lysed with RIPA buffer (1 mmol/L Halt Protease Inhibitor Single and 5 mmol/L EDTA). The concentration of each sample was determined using Bradford Protein Assay kit (ThermoFisher Scientific, Waltham, MA, USA) according to the manufacturer’s protocols.

### 4.8. Western Blotting

Protein lysates were made up at a final concentration of 5 µg/µL, including 4x NuPAGE^®^ LDS sample buffer (Life Technologies) and 10x sample-reducing agent (Life Technologies). Samples were then heated for 5 min at 95 °C, and 20 µg of protein was loaded onto 4–12% gradient NuPAGE Bis-Tris Protein gels. Proteins were separated for 55 min at 200 V in an XCell sure-lock tank with 1x NuPAGE^®^ MOPS SDS running buffer (Life Technologies). The Western transfer was then performed using a Bio-Rad TurboBlot in cold transfer buffer (25mM Tris-base, 75 mM Glycine, 10% Methanol, 0.1% SDS) and a 0.45 µm Immobilon-P PVDF membrane (Merck Millipore, Cork/IRL). Proteins were transferred for 90 min at 20 V with a 100 mA cut-off. PVDF membranes were blocked in 5% milk powder in TBS + Tween 0.1% (TBST) for 1 h. Blots were rinsed three times and washed for three times 5 min in TBST. Primary antibodies (1:2000) in TBST with 0.05% sodium azide were incubated with the blots over-night at 4 °C. Antibodies were from Santa Cruz Biotechnology, Heidelberg/DE: β-Actin sc-130656, GLUT2 sc-31826, Human SGLT3 sc-134521, rabbit iNOS C-19 sc-649. Blots were rinsed three times and washed three times for 5 min in TBST and blocked for 10 min. Anti-rabbit secondary antibody (sc-2004) (1:2000) in blocking buffer and blots were incubated for 1 h at room temperature. Blots were rinsed three times, washed three times for 10 min in TBST and incubated for 3 min with ECL Prime (GE Healthcare, Little Chalfont, UK). Protein signal was visualised using a SynGene G:BOX with the GeneSnap software, version 7.12 (Synaptics, Cambridge, UK) and a 1–20 min exposure time.

After visualization, the Restore™ Western Blot Stripping Buffer was added to strip the antibody membrane after incubation for 15 min at 37 °C. The membrane was washed twice with PBST and blocked for 60 min again for further incubating β-actin-primary antibody (Antibodies-online GmbH, Aachen, Germany), which was chosen to normalize the relative density.

### 4.9. Nitric-Oxide Detection

Cell media was harvested at day 21 from Caco-2/TC7 cells; 0.5 mL was diluted 1:3 in reaction buffer and filtered using Millipore 10,000 MWCO filters (Fisher Scientific, Leicestershire, UK) by centrifugation for 5 min with 14,000× *g* on an Eppendorf 5417R centrifuge. Filtrate was harvested and nitric oxide determined using a total Nitric Oxide detection kit following the supplied protocol (Enzo Lifesciences, Exeter, UK).

### 4.10. Lipids Analysis

Gas chromatography–mass spectrometry (GC) determination was performed to investigate the lipid composition of tongue, stomach, duodenum, jejunum and ileum samples. Samples were prepared as followed: the tissue lipid extracts were dried under nitrogen and reconstituted in 200 µL of toluene. Methanol (800 µL) was added to the lipid extract in toluene and then acidified with 100 µL of acetyl chloride to effect transesterification. Samples were incubated for 2 h at 60 °C, cooled and then neutralised with 3 mL of 6% *w*/*v* aqueous sodium carbonate. The resulting mixture was then centrifuged for 10 min at 2500× *g* at 40 °C to facilitate phase separation and the organic phase supernatant was transferred to a vial for GC analysis. GC analysis involves injection of 1 µL of the fatty acid methyl ester mixture into a gas chromatograph (Agilent Technologies 7890A) with the carrier gas flow (H2) set at 1 mL/min. Proportions of fatty acids as weight % were quantified by expressing the areas under chromatographic peaks as a % of the total integrated area. The proportion of oleic acid ethanolamide (OEA) and palmitoyl ethanolamide (PEA) in sample tissues was determined using liquid chromatography–mass spectrometry.

### 4.11. Statistical Analysis 

All data were shown in Microsoft Excel (2016). Gene expression levels were calculated with the Livak 2-△△CT method. Differences between two groups or among over three groups were analysed using *t*-test and one-way ANOVA, respectively, using SPSS 24.0 software (Chicago, IL, USA). Significant differences in statistic were accepted if *p* < 0.05, and highly significant differences in statistic were accepted if *p* < 0.01. All results are presented as mean ± standard error of the mean (SEM).

## Figures and Tables

**Figure 1 ijms-22-00137-f001:**
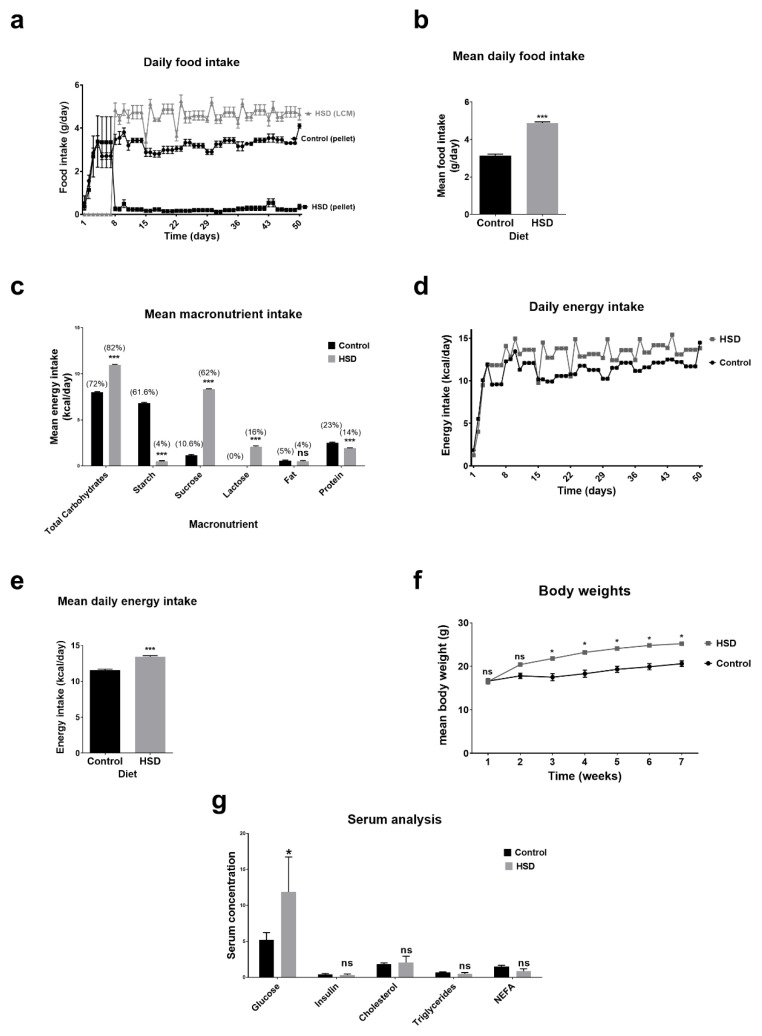
Food, macronutrient and energy intake, body weights and blood serum analysis of control fed and high sucrose diet (HSD) mice. (**a**) Daily food intake. (**b**) Mean daily food intake. (**c**) Macronutrient intakes calculated from the mean daily food intake (pellet and light condensed milk (LCM)) andthe macronutrient composition of control pellet chow and LCM (Materials and Methods: Table 3). (d) Daily energy intake. (**e**) Mean daily energy intake. (**f**) Body weights. (**g**) Blood serum analysis of glucose, cholesterol, triglycerides, NEFA (all in mmol/L) and Insulin (in ng/mL). Data are presented as means ± SE (*n* = 8). Statistical significance was tested with a Student’s *t*-test or one-way ANOVA. * *p* < 0.05, *** *p* < 0.001, ns = not significant.

**Figure 2 ijms-22-00137-f002:**
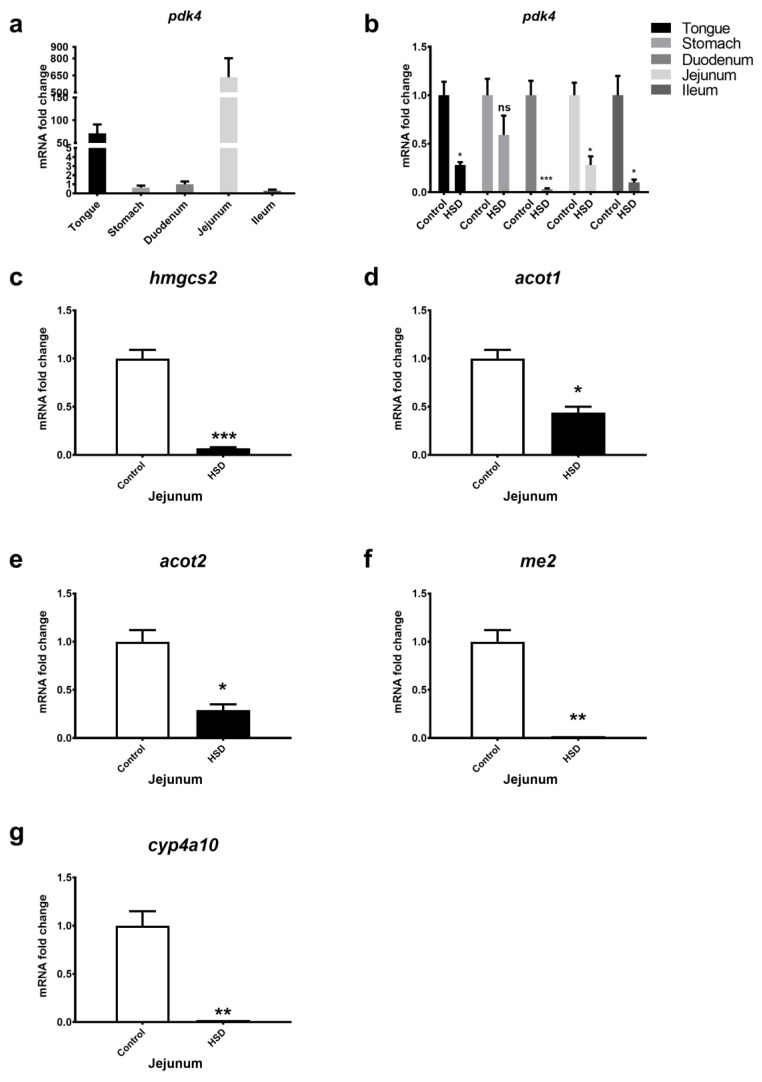
Fat oxidation gene expression levels in the gastrointestinal tract of mice. (**a**) Cephalocaudal gene expression of pdk4 in control fed mice. (**b**) pdk4 gene expression in gastrointestinal tract (GIT) segments obtained from mice fed a control diet vs. HSD. (**c**–**g**) hmgcs2, acot1, acot2, me1 and cyp4a10 gene expression levels in the jejunum of control and HSD fed mice. Values are normalised against three reference genes; relative expression was performed using the ddCT method and expressed relative to the control fed duodenum in panel a and control fed GIT segments in panel b, and the control fed jejunum in panels (**c**–**g**). Each bar represents the relative mean ± SE. * *p* < 0.05, ** *p* < 0.01, *** *p* < 0.001, ns = not significant.

**Figure 3 ijms-22-00137-f003:**
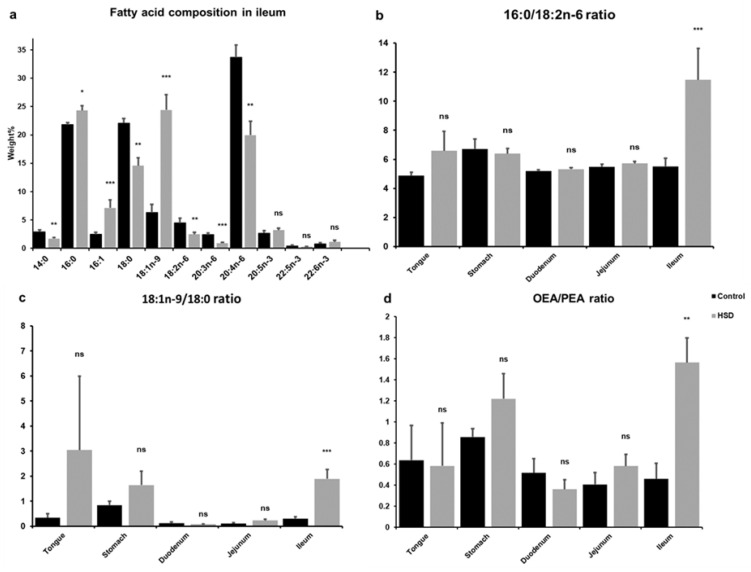
Fatty acid composition in the gastrointestinal tract of control and HSD mice (**a**) Fatty acid composition in ileum: 14:0: Myristic acid; 16:0: Palmitic acid; 16:1: Palmitoleic acid; 18:0: Stearic acid; 18:1n-9: Oleic acid; 18:2n-6: Linoleic acid; 20:3n-6: Dihomo-γ-linolenic acid; 20:4n-6: Arachidonic acid; 20:5n-3: Eicosapentaenoic acid; 22:5n-3: Docosapentaenoic acid; 22:6n-3: docosahexaenoic acid (**b**) 16:0/18:2n-6 ratio (**c**) 18:1n-9/18:0 ratio (**d**) Oleoylethanolamide/Palmitoylethanolamide (OEA/PEA) ratio in gastrointestinal segments obtained from control fed and HSD mice. Each bar represents mean ± SE. * *p* < 0.05; ** *p* < 0.01; *** *p* < 0.001; ns, no significant compared with control.

**Figure 4 ijms-22-00137-f004:**
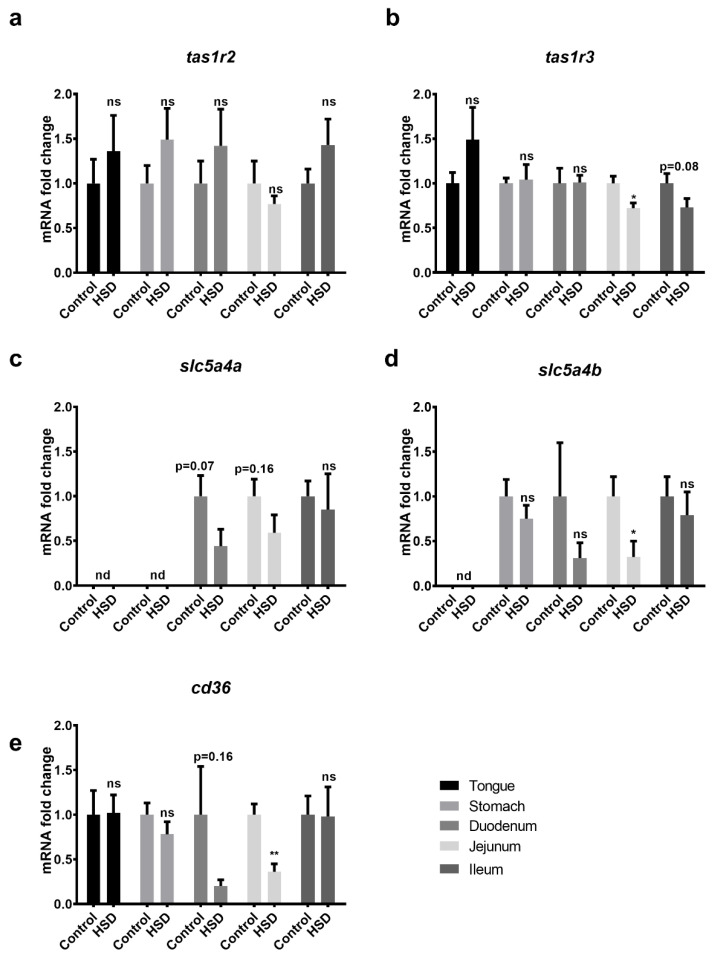
Nutrient sensor gene-expression in the gastrointestinal tract of control and HSD mice. Whole tongue, stomach, duodenum, jejunum and ileum were dissected from C57BL/6 mice fed either the HSD or the control diet. RNA was extracted for subsequent quantitative PCR of the sweet-taste and fat receptor subunits (**a**) tas1r2 (T1R2), (**b**) tas1r3 (T1R3), (**c**) slc5a4a (SGLT3a), (**d**) slc5a4b (SGLT3b) and (**e**) cd36. Values are normalised against three reference genes; relative quantitation was performed using the ddCT method and expressed relative to the control diet. Each bar represents the relative mean ± SE. * *p* < 0.05, ** *p* < 0.01, ns = not significant.

**Figure 5 ijms-22-00137-f005:**
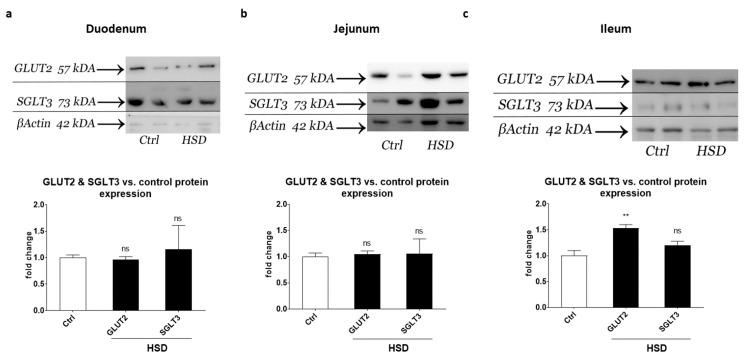
GLUT2 and SGLT3 protein expression levels in the small intestinal segments of mice. Whole (**a**) duodenum, (**b**) jejunum and (**c**) ileum were dissected from C57BL/6 mice fed a control diet or HSD (*n* = 8 per group) and proteins extracted for Western blots of β-Actin, GLUT2 and SGLT3. Representative blots are shown. Blots were quantified and normalised against β-Actin and are expressed relative to the control group. Each bar represents the relative mean ± SE. ** *p* < 0.01, ns = not significant.

**Figure 6 ijms-22-00137-f006:**
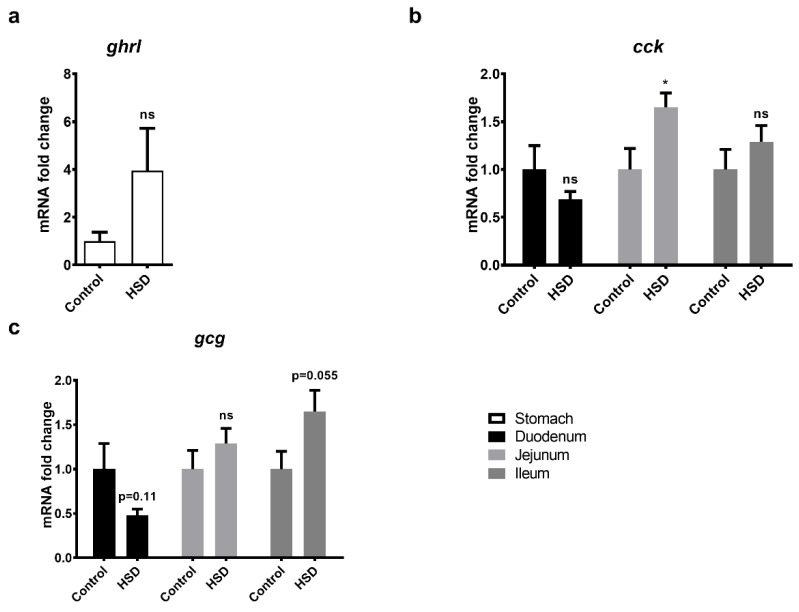
Gut peptide gene expression in the gastrointestinal tract of mice. Whole stomach, duodenum, jejunum and ileum were dissected from C57BL/6 mice fed the HSD or the control diet. RNA was extracted for subsequent quantitative PCR of the gut peptides (**a**) ghrl (Ghrelin), (**b**) cck (CCK) and (**c**) gcg (Preproglucagon). Values are normalised against three reference genes; relative expression was performed using the ddCT method and expressed relative to the control diet. Each bar represents the relative mean ± SE. * *p* < 0.05, ns = not significant.

**Figure 7 ijms-22-00137-f007:**
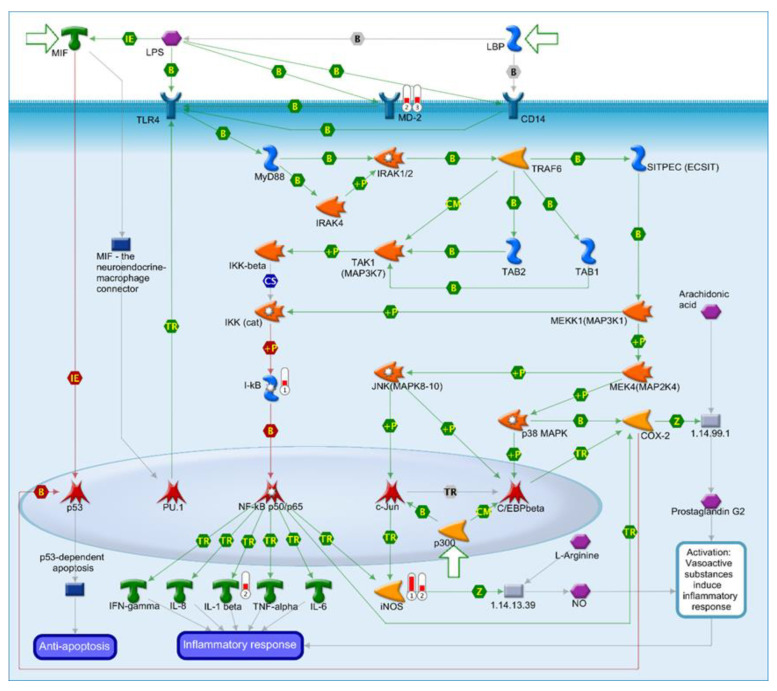
Biological network, containing lipopolysaccharide (LPS)-induced inflammation pathway generated by GeneGo MetaCore using the ”Analyze Pathways” algorithm. This pathway was generated from duodenal, jejunal and ileal gene expression data obtained from Affymetrix microarrays of pooled RNA samples of control and HSD mice groups. The threshold was set to a fold change of ≥1.5 (and ≤−1.5) to filter out genes whose expression was not markedly changed. Upregulated genes are marked with red thermometers; downregulated genes are marked with blue thermometers.

**Figure 8 ijms-22-00137-f008:**
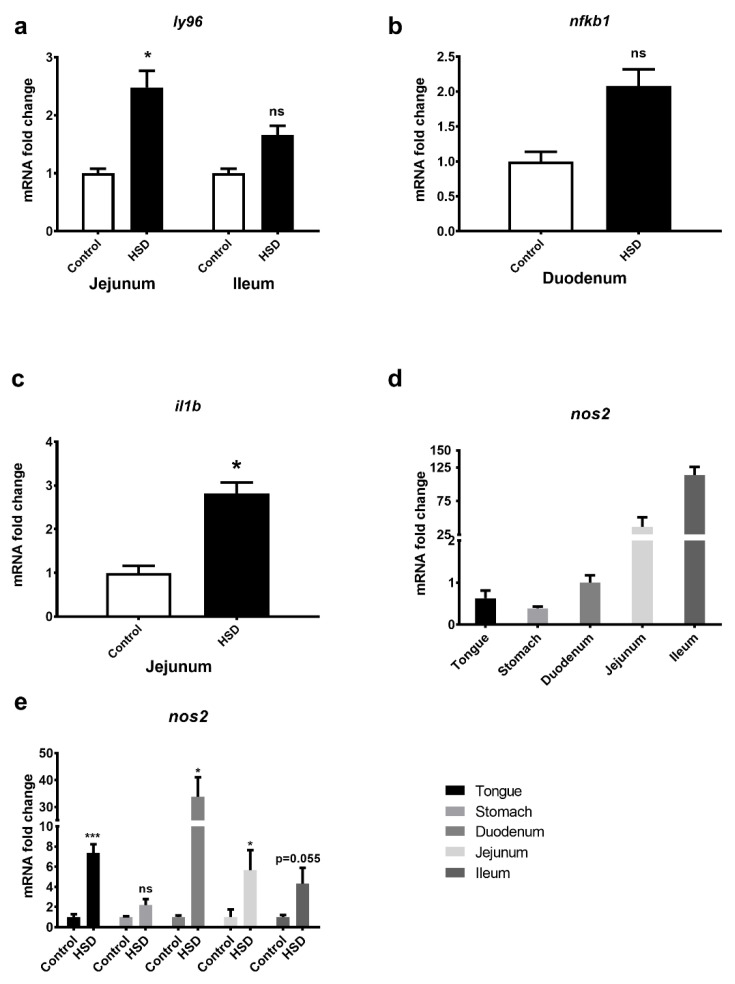
LPS-induced inflammation gene expression levels in the gastrointestinal tractof mice. Tongue, stomach, duodenum, jejunum and ileum were dissected from C57BL/6 mice fed either the control diet (*n* = 8) or HSD (*n* = 8) and RNA extracted for subsequent quantitative PCR. (**a**) LY96 expression in jejunum and ileum. (**b**) NFKβ1 in duodenum. (**c**) IL-1β in jejunum. (**d**) Cephalocaudal expression of NOS2 in control fed mice. (**e**) NOS2 expression in all GIT segments obtained from mice fed a control diet vs. HSD. Values are normalised against three reference genes; relative expression was performed using the ddCT method and expressed relative to duodenum for the cephalocaudal figures and the control diet for all other figures. Each bar represents the relative mean ± SE. * *p* < 0.05, *** *p* < 0.001, ns = not significant.

**Figure 9 ijms-22-00137-f009:**
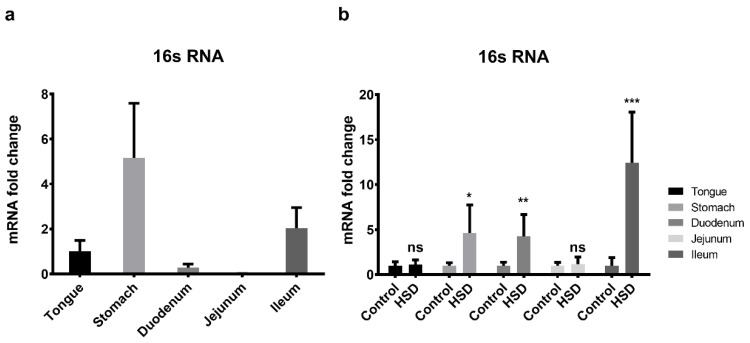
16S rRNA expression levels in the gastrointestinal tract of mice. (**a**) Cephalocaudal expression of 16S rRNA obtained from control fed mice. (**b**) 16S rRNA expression levels in gastrointestinal segments obtained from mice fed a control diet vs. HSD. Values are normalised against the housekeeping gene β-actin, and relative expression was performed using the ddCT method and expressed relative to the duodenum in panel a and control fed GIT segments in panel b. Results are expressed as means ± SE. HSD, high sucrose diets. * *p* < 0.05, ** *p* < 0.01, *** *p* < 0.001, ns = not significant.

**Figure 10 ijms-22-00137-f010:**
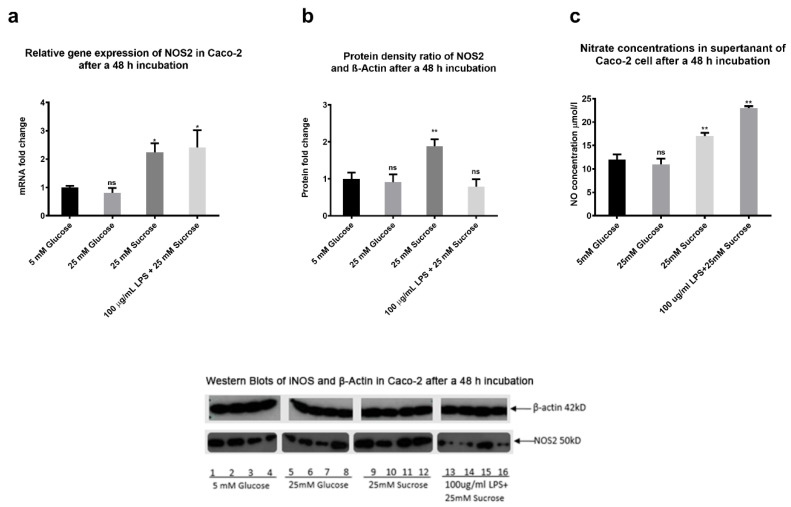
Nitric oxide studies in Caco-2 cells. Caco-2 cells were exposed to sugars and LPS for 48 h (*n* = 4 for each treatment). Total mRNA and proteins were then extracted and iNOS gene and protein expression measured by qPCR and Western blotting, respectively. Supernatants were also collected for NO gas analysis. (**a**) NOS2 gene expression levels normalized against β-actin. (**b**) iNOS protein expression levels. Protein density ratio of iNOS expression with β-actin normalization using Image-J software. (**c**) NO gas concentrations in the supernatants of Caco-2 cells. Values are presented as means ± SE. * *p* < 0.05, ** *p* < 0.01, ns = not significant. All P values versus 5 mM glucose group.

**Table 1 ijms-22-00137-t001:** Fold changes of regulated nutrient-sensor genes in the small intestine of HSD-fed mice compared with control fed mice. RNA was pooled from 8 mice per control and HSD group and analysed on an Affymetrix microarray chip.

	Protein	Gene Name	Fold Change Ctrl vs. HSD		
			Duo	Jej	Ile
**Sweet**	T1R2	tas1r2	1.15	−1.03	1.12
	T1R3	tas1r3	1.13	1.07	−1.05
	SGLT3a	slc5a4a	−2.61	−1.73	1.19
	SGLT3b	slc5a4b	−4.23	−3.0	−1.13
**Fat**	CD36/FAT	cd36	−4.62	−3.12	−1.41
	FFAR2	ffar2	1.02	1.04	1.09
	GPR120	gpr120	−1.10	−1.21	−1.13
**Umami**	mGluR4	grm4	−1.03	−1.13	−1.01
	T1R1	tas1r1	1.11	−1.07	−1.15
**Sour**	TRPP3	pkd2l1	1.15	1.00	−1.13
	ENaCα	scnn1a	−1.09	−1.15	−1.06
	ENaCβ	scnn1b	1.00	−1.31	−1.07
	ENaCγ	scnn1c	NA	NA	NA
**Bitter**	T2R102–T2R143	tas2r102–tas2r143	−1.43–1.33	−1.59–1.22	−1.27–1.06
**Signalling**	αGustducin	gnat3	1.28	1.31	1.29
	αTransducing	gnat1	1.19	−1.44	−1.74
	TRPM5	trpm5	1.66	1.20	1.14
	Phospholipaseβ2	plcb2	1.52	1.18	−1.04

**Table 2 ijms-22-00137-t002:** Key genes involved in LPS-induced inflammatory response.

**Key Gene**	**Protein Encoded**	**Fold Changes**	**Effect**
MD-2	Lymphocyte antigen 96	Duodenum: 1.17 Jejunum: 1.64 Ileum: 1.76	MD-2 associates with toll-like receptor 4 on the cell surface and responds to LPS, thus providing a link between the receptor and LPS signalling.
I-kB	Nuclear factor of kappa light polypeptide gene enhancer in B-cells inhibitor, epsilon (NFKBIE)	Duodenum: 1.63 Jejunum: 1.29 Ileum: 1.4	This protein complex controls DNA transcription, cytokine production and cell survival, playing a key role in mediating the immune response to infection.
IL-1β	Interleukin 1 beta	Duodenum: 1.22 Jejunum: 1.77 Ileum: 1.4	IL-1β is a member of the interleukin 1 cytokine family, an important regulator of the inflammatory response, and is involved in a variety of cellular activities, including cell proliferation, differentiation and apoptosis.
iNOS	Nitric oxide synthase 2	Duodenum: 4.74 Jejunum: 1.95 Ileum: 1.2	This protein is induced by a combination of lipopolysaccharide and certain cytokines, and NO gas, a mediator of several biological processes, is released.

**Table 3 ijms-22-00137-t003:** Macronutrient composition of control pellet chow and LCM.

	2% Corn Oil Chow		LCM + 1% Corn Oil	
	g/100 g	kcal/100 g	g/100 g	kcal/100 g
**Total carbohydrate**	68	255	59.5	223.1
**Starch**	58	217.5	0	0
**Sucrose**	10	37.5	47.3	177.4
**Lactose**	0	0	12.2	45.7
**Fat**	2	18	1.2	10.8
**Protein**	20	80	9.3	37.2
**Total**	**100**	**353**	**100**	**271.1**

**Table 4 ijms-22-00137-t004:** Primer sequences used in real-time RT-PCR.

Gene	Forward Primer: 5′-3′	Reverse Primer: 5′-3′
*18s*	gcaattattccccatgaacg	gggacttaatcaacgcaagc
*hprt1*	tcctcctcagaccgctttt	cctggttcatcatcgctaatc
*hmbs*	tccctgaaggatgtgcctac	aagggttttcccgtttgc
*slc5a1*	ctggcaggccgaagtatg	ttccaatgttactggcaaagag
*16s*	cgtgccagccgcggtaatacg	gggttgcgctcgttgcgggacttaacccaacat
*slc5a4a*	aaacccattcccgatgttc	tcgattctttcctccttactgttc
*slc5a4b*	ccgattcctgatgttcacct	atccgctcctctgtgttgtt
*gcg*	cacgcccttcaagacacag	gtcctcatgcgcttctgtc
*cck*	tgatttccccatccaaagc	gcttctgcagggactaccg
*ghrl*	ccagaggacagaggacaagc	catcgaagggagcattgaac
*tas1r2*	aagcatcgcctcctactcc	ggctggcaactcttagaacac
*tas1r3*	gaagcatccagatgacttca	gggaacagaaggacactgag
*cd36*	ttgtacctatactgtggctaaatgaga	cttgtgttttgaacatttctgctt
*ly96*	gccttctcagtcttggtggt	tctttccacggagattctgg
*nfkb1*	ctgacctgagcctctggac	gcaggctattgctcatcaca
*il1b*	gcccatcctctgtgactcat	aggccacaggtattttgtcg
*nos2*	ctttgccacggacgagac	tcattgtactctgagggctgac
*pdk4*	cgcttagtgaacactccttcg	cttctgggctcttctcatgg
*hmgcs2*	ctgtggcaatgctgatcg	tccatgtgagttcccctca
*acot1*	ggaggttggggaaaggtacaa	actccattcccagcccttga
*acot2*	tttctctgcggaaccgagg	tgctctcaggacagcgaaag
*me2*	gcagctcttcgaataaccaag	aagtgagcaatccccaagg

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
