# Peer review of "Chronic Effects of a High Sucrose Diet on Murine Gastrointestinal Nutrient Sensor Gene and Protein Expression Levels and Lipid Metabolism"

_ijms, 2020, doi:10.3390/ijms22010137_

Round 1

Reviewer 1 Report

This study aimed to identify changes in GIT function in mice chronically fed a high sucrose diet to induce obesity and diabetes

The investigators observed that dietary sucrose induced obesity/diabetes in mice is associated with alterations in numerous gastrointestinal genes involved in macronutrient sensing, appetite regulation, nutrient metabolism and intestinal microflora.

The study is interesting and significant and the data are presented in a clear and logical manner.

Minor revision:

Delete primers used from line 572 to 594.  Rather, provide a Table to list the primers used in this study

Author Response

Line 648 the primer sequences have now been included in a primer table 4

Reviewer 2 Report

The manuscript ijms-1020609 entitled "Chronic effects of a high sucrose diet on murine 2 gastrointestinal nutrient sensor gene and protein expression levels and lipid metabolism" is an original manuscript describing the role of chronic sucrose intake in changing gastrointestinal genes of nutrient sensing, inflammation and gut microbiota. Although having results of potential interest, the manuscript has several problems that should be addressed.

  • Some results lack novelty, as the effects of chronic sucrose intake in gastrointestinal inflammation and gut microbiota dysbiosis (PMID: 25762527; PMID: 29363272; PMID: 30230934). However, the theme is of relevance and important to study.
  • In the experimental protocol, it is questionable to house mice individually, which is a condition known to change food intake behaviour.
  • In diet design, why using LCM and not just supplement the standard diet with sucrose maintaining fats and changing starch for sucrose?
  • The manuscript has too many figures, which makes the manuscript difficult to read, for instance, Figures 1, 2, 3 and possibly 4 could be only one. And the same happens with others.
  • In figure 5, it is not clear what is the difference between A and B. The controls are the same? Is it a normalization? Nor the legend nor the figure (B) show which tissues is represented in each bar. The legend of Fig 5 is also poor. The difference between A and B is also not clear in Figure 13.
  • Figure 6 is not useful, because the diets have different lipid compositions.
  • In figure 9C, the protein levels are similar between controls and HSD, the differences are apparently due to the loading control.
  • Figure 10: authors only show data for ghrelin expression in the stomach but it is also expressed in the intestine.
  • Most of the figures have no significant differences. Authors should focus on significant results and try to understand what they mean. Otherwise, it is just a collection of data without a major conclusion.
  • The main problem of the manuscript is the lack of focus. Is it about nutrient-sensing, inflammation or microbiota? It shows data regarding a lot of mechanisms and is not conclusive about any of them. The conclusion also reveals this, because no conclusion is taken. The authors just say the diet-induced obesity is associated with gene expression differences, but they don’t mention them nor take any conclusion of that. The final sentence is clearly speculative, because none of the factors studied in the manuscript may serve as a therapeutic target in consequence of the results here presented.

Author Response

Reviewer 2: Some results lack novelty, as the effects of chronic sucrose intake in gastrointestinal inflammation and gut microbiota dysbiosis (PMID: 25762527 Improved Glucose Homeostasis in Obese Mice Treated With Resveratrol Is Associated With; PMID: 29363272 Microbiota in obesity: interactions with enteroendocrine, immune and central nervous systems; PMID: 30230934 Alterations in the Gut Microbiome: Microbiota modulation by eating patterns and diet composition: impact on food intake). However, the theme is of relevance and important to study.

Author response. Thank you for your comments. We have now included an additional statement and the reference by Mulders et al..

Line 78. Understanding the pathways by which dietary changes result in metabolic disease via alteration in gut microbiota, inflammation and nutrient sensing are therefore of interest (Mulders, de Git et al. 2018).

Reviewer 2: In the experimental protocol, it is questionable to house mice individually, which is a condition known to change food intake behaviour.

Author response: Mice were housed individually in order to capture food and calorie intake/day. Mice normally consume around 3-5 g of chow/day (12g chow /100 g body weight) and so the intakes of our control fed mice intake that were housed individually were within the normal range day.

Line 91: Adult mice normally consume 3-5 g of chow/day (12g chow /100 g body weight) 

Reviewer 2: In diet design, why using LCM and not just supplement the standard diet with sucrose maintaining fats and changing starch for sucrose?

Author response: The study was part of a much larger study looking at the protective effects of different oils in a mouse model of NASH/NAFLD, and manipulating the fat content of the LCM was a much more convenient vehicle for modulating fat intake than using solid chow control diets. A subgroup of the larger study was used in this report to identify changes in the GI tract gene, protein and metabolite levels following a switch to sucrose diet.

Reviewer 2: The manuscript has too many figures, which makes the manuscript difficult to read, for instance, Figures 1, 2, 3 and possibly 4 could be only one. And the same happens with others.

Author response: We accept this point and have combined Figs 1-4 into a multi panel, now Figures 1a-g and updated the text and figure legend ( Line 101). In addition, we have also combined the Western blot figure for GLUT2 and SGLT3, now Figures 5 a- c and updated the text and figure legend (Line 245).

Reviewer 2: In figure 5, it is not clear what is the difference between A and B. The controls are the same? Is it a normalization? Nor the legend nor the figure (B) show which tissues is represented in each bar. The legend of Fig 5 is also poor. The difference between A and B is also not clear in Figure 13.

Author response: Figure 5a (now Figure 2a) shows expression levels of the pdk4 gene in different regions of the GI tract obtained from control fed mice, and the data are normalized against duodenal pdk4 expression levels set at 100%. Figure 5b (Now 2b) shows changes in pdk4 gene expression levels in different regions of the GI tract obtained from control and HSD fed mice. All data are normalized against the labelled control fed intestinal segments set at 100%. The same applies to figure 13 (now figure 9). We have clarified this in the figures, which were missing the appropriate label, and also in the figure legends.

Lines 171…Figure 2. Fat oxidation gene expression levels in the gastrointestinal tract of mice. a) Cephalocaudal gene expression of pdk4 in control fed mice. b) pdk4 gene expression in GIT segments obtained from mice fed a control diet vs. HSD. c-g) hmgcs2, acot1, acot2, me1 and cyp4a10 gene expression levels in the jejunum of control and HSD fed mice. Values are normalised against three reference genes; relative expression was performed using the ddCT method and expressed relative to the control fed duodenum in panel a and control fed GIT segments in panel b, and the control fed jejunum in panels c-g. Each bar represents the relative mean ± SE. *P < 0.05, **P < 0.01, ***P < 0.001, ns = not significant

Lines 339 Figure 13 (now Figure 9). 16S rRNA expression levels in the gastrointestinal tract of mice a) Cephalocaudal expression of 16S rRNA in control fed mice. b) 16S rRNA expression level in GIT segments obtained from mice fed a control diet vs. HSD. Values are normalised against the housekeeping gene β-actin, and relative expression was performed using the ddCT method and expressed relative to the duodenum in panel a and control fed GIT segments in panel b. Results are expressed as means ± SE. HSD, high sucrose diets. *P < 0.05, **P < 0.01, ***P < 0.001, ns = not significant.

Reviewer 2: Figure 6 is not useful, because the diets have different lipid compositions.

Author response: It is true, the lipid compositions of the control (2% corn oil) and HSD (LCM 1% corn oil) diets are different. However, this was deliberate because pilot studies had shown the HSD mice had significantly increased LCM intake, resulting a different fat intake compared to controls. By adjusting the lipid composition levels of the control and LCM diet, the intake levels (and composition) of fat between the two study groups were statistically no different, as shown in figure 2. Changes in intestinal lipid metabolism are therefore most likely to be because of changes in sugar intakes. We have clarified this point in the paper.

Lines 560….In pilot studies HSD fed mice consumed large amounts of LCM and very little of the control chow diet resulting in changes in sugar as well as fat intakes. In order to ensure fat consumption (specifically linoleic acid) were similar between control and HSD mice, 1% corn oil was added to LCM and 2% corn oil to the control chow diet.

Reviewer 2: In figure 9C, the protein levels are similar between controls and HSD, the differences are apparently due to the loading control.

Authors response: Lane to lane variations in protein loading have been corrected for by normalizing the target protein expression levels against expression levels of beta actin. This is standard and the data shows that for most of the small intestinal segments, GLUT2 and SGLT3 expression levels are unchanged, despite gene expression levels changing.

Reviewer 2 : Figure 10: authors only show data for ghrelin expression in the stomach, but it is also expressed in the intestine.

Author response: Our control vs HSD gene array data set contains information on ghrelin’s expression levels in small intestine. However, ghrelin’s expression levels were much lower than for the incretins (…), and changes were minimal (duodenum -1.5, Jejunum 1.2 and ileum 1). These changes were below our chosen threshold (> or < 1.5) for further study. We have added a comment to this effect in the text.

Lines 277.. ..Array data indicated gene expression levels for ghrelin (duo:-1.52, jej: 1.03, ile:1.21) and gip (duo:-1.20, jej:,-1.16, ile:1.21) were minimally changed in all three small intestinal segments obtained from control and HSD fed mice (data not shown).

Reviewer 2: Most of the figures have no significant differences. Authors should focus on significant results and try to understand what they mean. Otherwise, it is just a collection of data without a major conclusion.

Authors response: We disagree with this point; all figures show significant data changes that we believe we have discussed thoroughly.  Trends are also presented as we believe higher powered studies would have shown statistical significance reinforcing the biological importance of the detected change (e.g ghrelin levels in stomach were four-fold higher in HSD mice). Data showing no changes were also presented for comparison (e.g. the putative sugar sensor SGLT3 was significantly downregulated by dietary sucrose, whereas the expression levels of the sweet taste receptor machinery (SGLT1 and t1r2/3) were unchanged.

Reviewer 2: The main problem of the manuscript is the lack of focus. Is it about nutrient-sensing, inflammation or microbiota? It shows data regarding a lot of mechanisms and is not conclusive about any of them. The conclusion also reveals this, because no conclusion is taken. The authors just say the diet-induced obesity is associated with gene expression differences, but they don’t mention them nor take any conclusion of that. The final sentence is clearly speculative, because none of the factors studied in the manuscript may serve as a therapeutic target in consequence of the results here presented.

Authors response: We have taken on board this criticism and have changed the conclusionary statements in the manuscript.

Line 538… In summary, dietary sucrose induced obesity/diabetes in mice is associated with alterations in numerous gastrointestinal genes involved in macronutrient sensing, appetite regulation, nutrient metabolism and intestinal microflora. Causality however remains an unanswered question. To better understand the importance of the changes in microflora and iNOS  in intestinal nutrient sensing and the HSD phenotype, in the future  a more detailed characterization of the microflora, probiotic and prebiotic interventions and treatment with the nitric oxide synthase inhibitor L-NG-nitro-arginine methyl ester) will be undertaken. These studies will identify viable therapeutic targets and strategies for the management of diet related metabolic disease.

Round 2

Reviewer 2 Report

The authors have responded appropriately to all the comments raised.